# Uhlmann phase of coherent states and the Uhlmann-Berry correspondence

Xin Wang[1], Xu-Yang Hou[1], Zheng Zhou[1], Hao Guo[1★] and Chih-Chun Chien[2†]

1 School of Physics, Southeast University, Jiulonghu Campus, Nanjing 211189, China
2 Department of physics, University of California, Merced, CA 95343, USA

★ guohao.ph@seu.edu.cn , † cchien5@ucmerced.edu

## Abstract

We first compare the geometric frameworks behind the Uhlmann and Berry phases in a fiber-bundle language and then evaluate the Uhlmann phases of bosonic and fermionic coherent states. The Uhlmann phases of both coherent states are shown to carry geometric information and decrease smoothly with temperature. Importantly, the Uhlmann phases approach the corresponding Berry phases as temperature decreases. Together with previous examples in the literature, we propose a correspondence between the Uhlmann and Berry phases in the zero-temperature limit as a general property except some special cases and present a conditional proof of the correspondence.

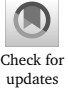

# 1   Introduction

The Berry phase [1] reveals geometric information of quantum wavefunctions via their phases acquired after an adiabatic cyclic process, and its concept has laid the foundation for understanding many topological properties of materials [2–13]. The theory of Berry phase is built on pure quantum states. For example, the ground state fits the description as the limit of a statistical ensemble at zero temperature. At finite temperatures, the density matrix describes thermal properties of a quantum system by associating a thermal distribution to all the states of the system. Therefore, it is an important task to generalize the Berry phase to the realm of mixed quantum states.

There have been several approaches to address this problem [14–21], among which the Uhlmann phase has attracted much attention recently since it has been shown to exhibit topological phase transitions at finite temperatures in several 1D, 2D, and spin-$j$ systems [22–26]. A key feature of those systems is the discontinuous jumps of the Uhlmann phase at the critical temperatures, signifying the changes of the underlying Uhlmann holonomy as the system traverses a loop in the parameter space. However, due to the complexity of the mathematical structure and physical interpretation, the knowledge of the Uhlmann phase is far less than that of the Berry phase in the literature. Moreover, only a handful of models allow analytical results of the Uhlmann phase to be obtained [22–30]. The Berry phase is purely geometric in the sense that it does not depend on any dynamical effect during the time evolution of the quantum system of interest [31]. Therefore, the theory of the Berry phase can be constructed in a purely mathematical manner. As a generalization, the Uhlmann phase of density matrices was built in an almost parallel way from a mathematical point of view and shares many geometric properties with the Berry phase. We will first summarize both the Berry and Uhlmann phases using a fiber-bundle language to highlight their geometric properties.

Next, we will present the analytic expressions of the Uhlmann phases of bosonic and fermionic coherent states and show that their values approach the corresponding Berry phases as temperature approaches zero. Both types of coherent states are useful in the construction of path integrals of quantum fields [32–37]. While any number of bosons are allowed in a single state, the Pauli exclusion principle restricts the fermion number of a single state to be zero or one. Therefore, complex numbers are used in the bosonic coherent states while Grassmann numbers are used in the fermionic coherent states. The bosonic coherent states are also used in quantum optics to describe radiation from a classical source [38–41]. Moreover, the Berry phases of coherent states can be found in the literature [42–45], and we summarize the results in Appendix A. Our exact results of the Uhlmann phases of bosonic and fermionic coherent states suggest that they indeed carry geometric information, as expected by the concept of holonomy and analogy to the Berry phase. We will show that the Uhlmann phases of both cases decrease smoothly with temperature without a finite-temperature transition, in contrast to some examples with finite-temperature transitions in previous studies [22–30]. As temperature drops to zero, the Uhlmann phases of bosonic and fermionic coherent state approach the corresponding Berry phases.

Our results of the coherent states, along with earlier observations [22, 24, 26], suggest the Uhlmann phase reduce to the corresponding Berry phase in the zero-temperature limit. The

correspondence is nontrivial because the Uhlmann phase requires full-rank density matrices, which cannot be satisfied only by the ground state at zero temperature. Moreover, the fiber bundle for density matrices in Uhlmann's theory is a trivial one [46], but the fiber bundle for wavevfunctions in the theory of Berry phase needs not be trivial. A similar question on why the Uhlmann phase agrees with the Berry phase in certain systems as temperature approaches zero was asked in Ref. [29] without an answer. In the last part of the paper, we present a detailed analysis of the Uhlmann phase at low temperatures to search for direct relevance with the Berry phase. With the clues from the previous examples, we present a conditional proof of the correspondence by focusing on systems allowing analytic treatments of the path-ordering operations.

Before showing the results, we present a brief comparison between the Uhlmann phase and another frequently mentioned geometrical phase for mixed quantum states proposed in Refs. [16, 47], which was originally introduced for unitary evolution but later extended to nonunitary evolution [48]. This geometrical phase was inspired by a generalization of the Mach-Zehnder interferometry in optics and was named accordingly as the interferometric phase. It has a different formalism with a more intuitive physical picture and has been measured in experiments [49]. In general situations, the interferometric phase can be expressed as the argument of a weighted sum of the Berry phase factors from each individual eigenstate. Thus, its relation to the Berry phase is obvious. However, the concise topological meaning of the interferometric phase is less transparent since it is not directly connected to the holonomy of the underlying bundle as the Uhlmann phase does. The reason has been discussed in a previous comparison [50] between the two geometrical phases. The interferometric phase relies solely on the evolution of the system state while the Uhlmann phase is influenced by the changes of both the system and ancilla, which result in the Uhlmann holonomy. Although Uhlmann's approach can be cast into a formalism parallel to that of the Berry phase as we will explain shortly, its exact connection to the Berry phase is still unclear. The Uhlmann-Berry correspondence discussed below will offer an insight into this challenging problem.

The rest of the paper is organized as follows. In Sec. 2, we first present concise frameworks based on geometry for the Berry and Uhlmann phases, using a fiber-bundle language. In Sec. 3, we derive the analytic expressions of the Uhlmann phases of bosonic and fermionic coherent states and analyze their temperature dependence. Additionally, the Uhlmann phase of a three-level system is also presented. Importantly, the Uhlmann phases of both types of coherent states and the three-level system are shown to approach the respective Berry phases as temperature approaches zero. In Sec. 4, we propose the generality of the correspondence between the Uhlmann and Berry phases in the zero-temperature limit and give a conditional proof. In Sec. 5, we discuss experimental implications and propose a protocol for simulating and measuring the Uhlmann phase of bosonic coherent states. Sec. 6 concludes out work. The Berry phases of bosonic and fermionic coherent sates and the special cases with a 1D Hilbert space are summarized in the Appendix.

## 2 Overview of Berry and Uhlmann phases

### 2.1 Berry phase in the bundle language

We adopt the natural units with $k_B = 1 = \hbar$. The first part of the overview of the Berry phase follows Refs. [1, 31, 46]. The Berry phase arises under a cyclic adiabatic evolution experienced by a quantum state through external parameters. The Hamiltonian of the system is given by $\hat{H}(\mathbf{R})$, where $\mathbf{R} = (R_1, R_2, \cdots, R_k)^T \in M$ is the collection of the external parameters. If the state $|n(\mathbf{R}(t))\rangle$ evolves adiabatically along a closed curve $C(t) := \mathbf{R}(t)$ ($0 \le t \le \tau$) in the

parameter space $M$, at the end of the evolution the final state obtains a geometric phase

$$\theta_n = \mathrm{i} \int_0^\tau \mathrm{d}t \langle n(\mathbf{R}(t)) | \frac{\mathrm{d}}{\mathrm{d}t} | n(\mathbf{R}(t)) \rangle, \tag{1}$$

with respect to the initial state.

The theory of Berry phase can be cast into another equivalent formalism by introducing the parallel-transport of quantum states. If two pure states $|\psi_{1,2}\rangle$ are in phase with each other, i.e. $\arg\langle\psi_1|\psi_2\rangle = 0$ or $\langle\psi_1|\psi_2\rangle = \langle\psi_2|\psi_1\rangle > 0$, they are also said to be parallel with each other. Thus, the parallel-transport of a state $|\psi(t)\rangle$ is defined via

$$\langle\psi(t)|\psi(t+\mathrm{d}t)\rangle = \langle\psi(t+\mathrm{d}t)|\psi(t)\rangle > 0, \tag{2}$$

whose differential form is

$$\langle\psi(t)|\frac{\mathrm{d}}{\mathrm{d}t}|\psi(t)\rangle = 0. \tag{3}$$

The parallel condition lacks transitivity, so it does not define an equivalence relation. Therefore, even if a system follows parallel transport, its quantum state, say $|n(\mathbf{R}(t))\rangle$, may gradually acquire an extra phase other than the dynamical phase. We assume $|\psi(t)\rangle = \mathrm{e}^{\mathrm{i}\theta_n(t)}|n(\mathbf{R}(t))\rangle$ and substitute it into the condition (3) to get

$$\mathrm{i}\frac{\mathrm{d}\theta_n}{\mathrm{d}t} + \langle n(\mathbf{R}(\mathbf{t}))|\frac{\mathrm{d}}{\mathrm{d}t}|n(\mathbf{R}(\mathbf{t}))\rangle = 0. \tag{4}$$

Solving this differential equation, we directly obtain the Berry phase shown in Eq. (1). Using $\frac{\mathrm{d}}{\mathrm{d}t} = \dot{\mathbf{R}} \cdot \nabla_{\mathbf{R}}$, it can be also expressed as

$$\theta_n = \arg\langle\psi(0)|\psi(\tau)\rangle = \mathrm{i}\oint_C \mathrm{d}t \langle n(\mathbf{R}(t))|\nabla_{\mathbf{R}}|n(\mathbf{R}(t))\rangle \cdot \mathrm{d}\mathbf{R}, \tag{5}$$

which carries geometric information of $C(t)$ in the parameter space. Accordingly, the Berry phase is a geometric phase that a quantum state obtains after being parallel-transported along a loop in the parameter space. This means that the Berry phase factor $\mathrm{e}^{\mathrm{i}\theta_n}$ is actually a holonomy in the language of differential geometry. Based on these discussions, the theory of Berry phase can be elegantly illustrated in a principle-bundle description. Some details can be found in Ref. [31], and here we present an improved and simplified discussion.

During an adiabatic evolution, no energy-level crossing occurs. Thus, once a quantum system initially starts from the $n$th level $|n(\mathbf{R}(0))\rangle$, it will stay in the instantaneous state $|n(\mathbf{R}(t))\rangle$. Hence, we will use the abbreviation $|\mathbf{R}\rangle \equiv |n(\mathbf{R})\rangle$ hereafter. Define $\mathbb{P} = \{|\mathbf{R}\rangle | \langle\mathbf{R}|\mathbf{R}\rangle = 1\}$. Since $|\mathbf{R}\rangle \sim \mathrm{e}^{\mathrm{i}\chi}|\mathbf{R}\rangle$ where $\chi$ is an arbitrary phase, the genuine phase space of the system is $H = \mathbb{P}/\sim$. We construct a fiber bundle $P(H, \mathrm{U}(1))$, where $P$ is the total space, $H$ is the base manifold and $\mathrm{U}(1)$ is the structure group. A projective operator $\pi : P \to H$ acts as $\pi(\mathrm{e}^{\mathrm{i}\chi}|\mathbf{R}\rangle) = |\mathbf{R}\rangle, \forall \mathrm{e}^{\mathrm{i}\chi} \in \mathrm{U}(1)$. Conversely,

$$\pi^{-1}(|\mathbf{R}\rangle) = \{g|\mathbf{R}\rangle | g \in \mathrm{U}(1)\} \tag{6}$$

is the fiber $F_{\mathbf{R}}$ at the point $|\mathbf{R}\rangle$, which is isomorphic to $\mathrm{U}(1)$. Thus, what we construct is a $\mathrm{U}(1)$-principle bundle. A section $\sigma : H \to P$ is a smooth map such that $\pi \circ \sigma = 1_H$, which locally fixes the phase of $|\mathbf{R}\rangle$ as $\sigma(|\mathbf{R}\rangle) = \mathrm{e}^{\mathrm{i}\theta(\mathbf{R})}|\mathbf{R}\rangle$.

The loop $C(t)$ induces a loop in $H$ as $\gamma(t) := |\mathbf{R}(t)\rangle$ ($|\mathbf{R}(0)\rangle = |\mathbf{R}(\tau)\rangle$). A curve $\tilde{\gamma}(t) \in P$ is called a lift of $\gamma(t)$ if $\pi \circ \tilde{\gamma} = \gamma$. The formerly mentioned $|\psi(t)\rangle = \mathrm{e}^{\mathrm{i}\theta_n(t)}|n(\mathbf{R}(t))\rangle$ is actually a

lift of $\gamma$. Let $X$ and $\tilde{X}$ be the tangent vectors to $\gamma$ and $\tilde{\gamma}$, respectively, then they satisfy $\pi_* \tilde{X} = X$. Moreover, we introduce a connection 1-form at $|\psi\rangle$ as

$$\omega_{|\psi\rangle} = \langle\psi|\mathrm{d}_P|\psi\rangle\,, \tag{7}$$

where $\mathrm{d}_P$ is the exterior derivative on $P$. Note $\tilde{X}$ can be locally expressed as $\tilde{X} = \frac{\mathrm{d}}{\mathrm{d}t}$ since $\tilde{\gamma}$ is parameterized by $t$. Then Eq. (4) can be written in the more generic form

$$\omega(\tilde{X}) = 0\,, \tag{8}$$

which is equivalent to the parallel-transport condition (3). This indicates that $\tilde{X}$ is a horizontal vector belonging to $TP$. Here $TP$ is the tangent bundle of $P$. Accordingly, $\tilde{\gamma}(t)$ is called the horizontal lift of $\gamma(t)$. The pullback of $\omega$ by $\sigma$ is $A_B = \sigma^*\omega = \langle\psi|\mathrm{d}_H|\psi\rangle$, where $\mathrm{d}_H$ is the exterior derivative on $H$. Since $\mathrm{d}_H$ does not act on the fiber space, $A_B$ is also expressed as

$$A_B = \langle\mathbf{R}|e^{-\mathrm{i}\theta_n}\mathrm{d}_H\left(e^{\mathrm{i}\theta_n}|\mathbf{R}\rangle\right) = \langle\mathbf{R}|\mathrm{d}_H|\mathbf{R}\rangle\,, \tag{9}$$

i.e. it is the well-known Berry connection on the base manifold $H$. Let $g(t) = e^{\mathrm{i}\theta_n(t)}$. $\omega$ can be conversely constructed as

$$\omega = \pi^*A_B + g^{-1}\mathrm{d}_P g\,. \tag{10}$$

A connection defined by Eq. (10) is also called an Ehresmann connection [51]. Using this, the condition (8) becomes

$$0 = \pi^*A_B(\tilde{X}) + g^{-1}\mathrm{d}_P g(\tilde{X}) = A_B(\pi_*\tilde{X}) + g^{-1}\frac{\mathrm{d}g}{\mathrm{d}t}\,, \tag{11}$$

which is equivalent to

$$\nabla_X g = 0\,. \tag{12}$$

Here $\nabla_i = \frac{\partial}{\partial R_i} + A_{Bi}$ is the covariant derivative associated with the Berry connection. Hence, the parallel-transport condition indicates that the phase factor, viewed as a vector in the fiber space, is parallel transported along $\gamma(t) \in H$ (or equivalently, $C(t) \in M$). Thus, $g(\tau) = e^{-\oint_C A_B}$ is a holonomy of the bundle, called the Berry holonomy. The Berry phase $\theta_B = \arg g(\tau)$ is a measure of the loss of parallelity after the system is parallel-transported along a loop.

There are more features in the fiber bundle. According to Eq. (8), the Ehresmann connection $\omega$ naturally separates $TP$ into the horizontal and vertical subspaces as $TP = HP \oplus VP$. It is also worthwhile to calculate $\omega(\tilde{X}^V)$, where $\tilde{X}^V \in VP$ is a vertical vector. Let $u(t) = \omega(\tilde{X}^V)$. Since $\tilde{X}^V$ is vertical, it follows that $\pi_*\tilde{X}^V = 0$. Following a similar derivation as Eq. (11), we get

$$u(t) = \pi^*A_B(\tilde{X}^V) + g^{-1}\mathrm{d}_P g(\tilde{X}^V) = g^{-1}\frac{\mathrm{d}g}{\mathrm{d}t}\,, \tag{13}$$

which further implies

$$g(t) = e^{\int_0^t u(t')\mathrm{d}t'} g(0)\,. \tag{14}$$

Here $e^{\int_0^t u(t')\mathrm{d}t'}$ is a phase transformation induced by a curve in the fibre space, and $u \in \mathfrak{u}(1)$ is its generator. Moreover, $\tilde{X}^V$ is the tangent vector of the curve $e^{\int_0^t u(t')\mathrm{d}t'}$, and we follow the terminology of Ref. [51] to write $\tilde{X}^V = u^\#$. Consequently, we have

$$\omega(u^\#) = u \tag{15}$$

if $u^\#$ is a vertical vector. We emphasize that the generalizations of Eqs. (8) and (15) play important roles in the theory of Uhlmann phase.

## 2.2 Uhlmann phase in the bundle language

A generalization of the Berry phase to mixed states is both natural and necessary, given the abundance of phenomena in nature described by mixed states. However, mixed quantum states are usually represented by density matrices, which are Hermitian operators and carry no explicit information about phase. Inspired by the structure $\rho = |\psi\rangle\langle\psi|$ for the density matrix of a pure state, Uhlmann introduced [14] the decomposition $\rho = WW^\dagger$ for a generic full-rank density matrix $\rho$, where $W$ is called the purification or amplitude of $\rho$. The decomposition is not unique because $W = \sqrt{\rho}U$ with $U \in \mathrm{U}(N)$ also satisfies the decomposition. Here $N$ is the dimension of the Hilbert space, and $U$ is called the phase factor of $W$. One may see the analogy of a pure-state wavefunction: $\psi(\mathbf{x}) = \sqrt{|\psi(\mathbf{x})|^2}e^{i\arg\psi(\mathbf{x})}$. If $\rho$ is diagonalized as $\rho = \sum_n \lambda_n |n\rangle\langle n|$, the purification is accordingly expressed as $W = \sum_n \sqrt{\lambda_n}|n\rangle\langle n|U$. Importantly, there is a corresponding state-vector representation $|W\rangle = \sum_n \sqrt{\lambda_n}|n\rangle \otimes U^T|n\rangle$, called the purified state of $\rho$. The inner product of two purified states is the Hilbert-Schmidt product between two purifications:

$$\langle W_1|W_2\rangle = \mathrm{Tr}(W_1^\dagger W_2)\,. \tag{16}$$

A key point in the construction of the theory of Uhlmann phase is to extend the parallel-transport condition (3) to mixed states. A direct and naive generalization seems to be

$$\langle W(t)|\frac{\mathrm{d}}{\mathrm{d}t}|W(t)\rangle = 0\,. \tag{17}$$

However, this only leads to a single equation and cannot determine the $N \times N$ matrix $W$. On the other hand, it can be found that the Fubini-Study length along a curve $C(t)$, $L_{\mathrm{FS}} = \int_{C,0}^\tau \sqrt{\langle\dot\psi|\dot\psi\rangle}\mathrm{d}t$, is minimized if and only if Eq. (3) holds [52,53]. A similar result holds for mixed states: The Hilbert-Schmidt length $L_{\mathrm{HS}} = \int_{C,0}^\tau \sqrt{\mathrm{Tr}(\dot W^\dagger \dot W)}\mathrm{d}t$ is minimized if and only if [28,53]

$$\dot W W^\dagger = W^\dagger \dot W\,, \tag{18}$$

which implies $\mathrm{Im}\langle W(t)|\frac{\mathrm{d}}{\mathrm{d}t}|W(t)\rangle = 0$. Eq. (17) can be deduced from this condition by noting that $\langle W(t)|W(t)\rangle = 1$. The matrix equation (18) has $N \times N$ entries, giving $N \times N$ restrictions. Hence, the condition is much stronger than Eq. (17).

The Uhlmann phase was introduced from a purely mathematical manner, and its physical interpretation still needs more work. Following the geometric description of the Berry phase, we first construct a $\mathrm{U}(N)$-principle bundle $P(H, \mathrm{U}(N))$ for mixed states, where $H$ is the base manifold including all $N$-dimensional full-rank density matrices, $P$ is the total space spanned by $W$, and a projection $\pi : P \to H$ is defined by

$$\pi(W) = WW^\dagger = \rho\,. \tag{19}$$

Here $\mathrm{U}(N)$ is the structure group, which contains all unitary phase-factor transformations. Conversely, a smooth map $\sigma : H \to P$ satisfying $\pi \circ \sigma = 1_P$ is called a section. There is a global section $\sigma(\rho) = \sqrt{\rho}$ defined on the entire $H$. Thus, this principle bundle is always trivial [46]. Nevertheless, many interesting and instructive results can still be inferred from the formalism, as we will show below.

When the system traverses a closed curve $C(t) := \mathbf{R}(t) \in M$ ($0 \le t \le \tau$), the density matrix evolves along an induced loop $\gamma(t) := \rho(t) \equiv \rho(\mathbf{R}(t))$ in $H$ accordingly. Similar to

the geometric description of the Berry phase, we set to find a horizontal lift $\tilde{\gamma}$ of $\gamma$ such that when the corresponding purification varies along $\tilde{\gamma}$, the parallel-transport condition (18) is satisfied. This requirement can be fulfilled if a connection $\omega$ defined on $P$ meets the condition $\omega(\tilde{X}) = 0$, where $\tilde{\gamma}$ is the tangent vector of $\tilde{X}$. To find $\omega$, we return to the parallel-transport condition (18), which can be rewritten as

$$W^{\dagger}\mathrm{d}_P \dot{W}(\tilde{X}) - \mathrm{d}_P W(\tilde{X})W^{\dagger} = 0. \tag{20}$$

A trial form of $\omega$ is $\omega = W^{\dagger}\mathrm{d}_P \dot{W} - \mathrm{d}_P W W^{\dagger}$. However, this does not meet the proper definition for a connection. It can be shown that $\omega$ defined this way does not transform like a gauge potential under a gauge transformation $W' \to WV$, where $V \in \mathrm{U}(N)$. To resolve the problem, we make use of Eq. (15) and note that a curve in the fiber space $\pi^{-1}(\rho)$ can always be expressed as $W(t) = \sqrt{\rho}\mathrm{e}^{tu}$, where $u \in \mathfrak{u}(N)$ is an anti-Hermitian matrix. Let $\tilde{X}^V$ be the tangent vector of this curve, which is by definition a vertical vector. It is straightforward to find

$$\mathrm{d}_P W(\tilde{X}^V) = Wu. \tag{21}$$

Thus, by replacing the horizontal vector in the left-hand-side of Eq. (20) by $\tilde{X}^V$ and using $u^{\dagger} = -u$, we get

$$W^{\dagger}\mathrm{d}_P \dot{W}(\tilde{X}^V) - \mathrm{d}_P W(\tilde{X}^V)W^{\dagger} = W^{\dagger}Wu - uW^{\dagger}W. \tag{22}$$

Moreover, since $u$ is the generator of the curve $W(t) = \sqrt{\rho}\mathrm{e}^{tu}$, whose tangent vector is $\tilde{X}^V$, we can also write $\tilde{X}^V = u^{\#}$ as before. A generalization of Eq. (15) is $\omega(\tilde{X}^V) = u$. Substituting this into the right-hand-side of Eq. (22), we have

$$W^{\dagger}\mathrm{d}_P \dot{W}(\tilde{X}^V) - \mathrm{d}_P W(\tilde{X}^V)W^{\dagger} = W^{\dagger}W\omega(\tilde{X}^V) - \omega(\tilde{X}^V)W^{\dagger}W. \tag{23}$$

The identity holds even for a horizontal vector $\tilde{X}^H$ due to Eq. (20) and $\omega(\tilde{X}^H) = 0$. Thus, the connection $\omega$ satisfies the following equation

$$W^{\dagger}\mathrm{d}_P W - \mathrm{d}_P W W^{\dagger} = W^{\dagger}W\omega - \omega W^{\dagger}W. \tag{24}$$

It can be verified that under a gauge transformation $W' \to WV$, $\omega$ defined by Eq. (24) transforms as $\omega' = V^{\dagger}\omega V + V^{\dagger}\mathrm{d}_P V$ and qualifies as a non-Abelian gauge potential. In Uhlmann's original paper [15], Eq. (24) is introduced as an ansartz to define a connection over the whole bundle. Here we find that it can be directly obtained from the condition $\omega(u^{\#}) = u$.

The pullback of $\omega$ by $\sigma$ is the Uhlmann connection $A_U = \sigma^*\omega$. Let $U = \mathrm{e}^{tu}$, and we have $\omega(\tilde{X}^V) = u = U^{\dagger}\frac{\mathrm{d}U}{\mathrm{d}t}$. Based on these results and $\omega(\tilde{X}^H) = 0$, if $\omega$ is the Ehresmann connection, it can be expressed as

$$\omega = U^{\dagger}\pi^* A_U U + U^{\dagger}\mathrm{d}_P U, \tag{25}$$

which is the non-Abelian generalization of Eq. (10). Moreover, contracting both sides of Eq. (25) with a horizontal vector $\tilde{X}$ leads to $A_U(X) = -\frac{\mathrm{d}U}{\mathrm{d}t}U^{\dagger}$, or equivalently,

$$\nabla_X U = \frac{\mathrm{d}U}{\mathrm{d}t} + A_U(X)U = 0. \tag{26}$$

Here $X = \pi_* \tilde{X}$ is the tangent vector to $\gamma$. Similarly, the equation shows that the phase factor $U$ is parallel-transported along the loop $\gamma$. Solving the equation, we get

$$U(\tau) = \mathcal{P}\mathrm{e}^{-\oint_C A_U}U(0), \tag{27}$$

where $\mathcal{P}$ is the path-ordering operator. Note $\mathcal{P}e^{-\oint_C A_U}$ is the Uhlmann holonomy, and the Uhlmann phase is

$$\theta_U = \arg\langle W(0)|W(\tau)\rangle = \arg \operatorname{Tr}\left[\rho(0)\mathcal{P}e^{-\oint_C A_U}\right]. \tag{28}$$

To derive an explicit expression of $A_U$, we plug $W = \sqrt{\rho}U$ into Eq. (24) and obtain

$$U^\dagger[\sqrt{\rho}, \mathrm{d}_P\sqrt{\rho}]U + U^\dagger\rho\mathrm{d}_P U + U^\dagger\mathrm{d}_P U U^\dagger\rho U = U^\dagger\rho U\omega + \omega U^\dagger\rho U. \tag{29}$$

Next, we use Eq. (25) to get

$$\rho\pi^*A_U + \pi^*A_U\rho = -[\mathrm{d}_P\sqrt{\rho},\rho]. \tag{30}$$

When restricted on $H$, it reduces to

$$\rho A_U + A_U\rho = -[\mathrm{d}_H\sqrt{\rho},\rho]. \tag{31}$$

Evaluating the matrix elements of both sides in the eigenstates of $\rho$, we get

$$A_U = -\sum_{n,m=1}^{N} |n\rangle \frac{\langle n|[\mathrm{d}\sqrt{\rho}, \sqrt{\rho}]|m\rangle}{\lambda_n + \lambda_m}\langle m|, \tag{32}$$

where we have omitted the subscript $H$ for convenience. We note that only when $N > 1$, $A_U$ may be nonzero since the representation of a commutator is trivial in a 1D Hilbert space (see Appendix. B for details).

We further simplify the expression (32) of $A_U$, which will be useful in our latter discussion on the similarity with the Berry connection $A_B$. Using $\sqrt{\rho} = \sum_n \sqrt{\lambda_n}|n\rangle\langle n|$, we have

$$[\sqrt{\rho}, \mathrm{d}\sqrt{\rho}] = \sum_n \lambda_n (|n\rangle\mathrm{d}\langle n| - \mathrm{d}|n\rangle\langle n|) + \sum_{nm}\sqrt{\lambda_n\lambda_m}(|n\rangle\langle n|\mathrm{d}|m\rangle\langle m| - |m\rangle(\mathrm{d}\langle m|)|n\rangle\langle n|). \tag{33}$$

By interchanging the indices $n \leftrightarrow m$ in the last term and using $(\mathrm{d}\langle n|)|m\rangle = -\langle n|\mathrm{d}|m\rangle$, it becomes

$$[\sqrt{\rho}, \mathrm{d}\sqrt{\rho}] = -\sum_{nm}\left(\sqrt{\lambda_n} - \sqrt{\lambda_m}\right)^2 |n\rangle\langle n|\mathrm{d}|m\rangle\langle m|, \tag{34}$$

and the Uhlmann connection becomes

$$A_U = -\sum_{n\neq m}\frac{\left(\sqrt{\lambda_n} - \sqrt{\lambda_m}\right)^2}{\lambda_n + \lambda_m}|n\rangle\langle n|\mathrm{d}|m\rangle\langle m|. \tag{35}$$

## 3 Uhlmann phase of coherent states

Here we apply the framework to find the Uhlmann phases of bosonic and fermionic harmonic oscillators. The corresponding Berry phases are summarized in Appendix A.

### 3.1 Bosonic coherent state

Here we evaluate the Uhlmann phase of bosonic coherent states, which may be constructed from bosonic harmonic oscillators [35,39]. The Hamiltonian of a single harmonic oscillator is $\hat{H} = \hbar\omega(a^\dagger a + \frac{1}{2})$, where $a, a^\dagger$ are the annihilation and creation operators satisfying $[a, a^\dagger] = 1$.

The energy levels of system are characterized by $\hat{H}|n\rangle = \hbar\omega(n+\frac{1}{2})|n\rangle$ with $n = 0, 1, 2, \cdots$. Previously studied examples of the Uhlmann phase of low-dimensional systems [22, 26, 28] and spin-$j$ systems [24, 25] are both in finite-dimensional Hilbert spaces. The bosonic harmonic oscillator will give an infinite-dimensional example. The parallel transport of a canonical ensemble of harmonic oscillators can be realized with the help of coherent states defined by operating the translation operator on the ground state: $|z\rangle = D(z)|0\rangle \equiv e^{za^\dagger - \bar{z}a}|0\rangle$. Here $D(z)$ satisfies

$$D(z)aD^\dagger(z) = a - z, \quad D(z)a^\dagger D^\dagger(z) = a^\dagger - \bar{z}. \tag{36}$$

Moreover, $|z\rangle$ is the ground state of the translated Hamiltonian $\hat{H}(z) = D(z)\hat{H}D^\dagger(z)$. The excited states are obtained in a similar manner: $|n, z\rangle = D(z)|n\rangle$, $n \geq 1$.

The parameter space is thus identified as the complex $z$ plane, and a loop for generating the holonomy may be chosen as $C(t) := z(t)$ with $z(0) = z(\tau)$ ($0 \leq t \leq \tau$). Our convention is that the counterclockwise direction of $C(t)$ follows the increase of $t$. The continuous transformation $D(z(t))$ generates an induced loop $\gamma(t) := \rho(z(t))$ in the manifold of density matrices, where

$$\rho(z) = \frac{1}{Z}e^{-\beta\hat{H}(z)} = D(z)\rho(0)D^\dagger(z). \tag{37}$$

Here $\rho(0) = \frac{1}{Z}e^{-\beta\hat{H}}$. Since $D(z)$ is unitary, the eigenvalues of $\rho$ are invariant under the action of $D(z)$, given by $\lambda_n = \frac{1}{Z}e^{-\beta\hbar\omega(n+\frac{1}{2})}$. Decomposing the density matrix, one obtains the purification $W(z(t)) = \sqrt{\rho(z(t))}U(z(t))$. As long as the phase factor $U(t) \equiv U(z(t))$ satisfies the parallel-transport equation (26) along $\gamma(t)$ (or $C(t)$ equivalently), the final state will acquire an Uhlmann phase relative to the initial state.

Using Eq. (35), the Uhlmann connection is given by

$$\begin{aligned}
A_U &= -\sum_{n \neq m} \frac{(\sqrt{\lambda_n} - \sqrt{\lambda_m})^2}{\lambda_n + \lambda_n}|n, z\rangle\langle n, z|\mathrm{d}|m, z\rangle\langle m, z| \\
&= -\sum_{n \neq m} \chi_{nm}D(z)|n\rangle\langle n|D^\dagger(z)\mathrm{d}D(z)|m\rangle\langle m|D^\dagger(z),
\end{aligned} \tag{38}$$

where $\chi_{nm} = \frac{(e^{-\frac{n}{2}\beta\hbar\omega} - e^{-\frac{m}{2}\beta\hbar\omega})^2}{e^{-n\beta\hbar\omega} + e^{-m\beta\hbar\omega}}$. It can be shown that

$$D^\dagger(z)\mathrm{d}D(z) = \left(a^\dagger + \frac{1}{2}\bar{z}\right)\mathrm{d}z - \left(a + \frac{1}{2}z\right)\mathrm{d}\bar{z}. \tag{39}$$

Using the above equation and $\langle n|a^\dagger = \sqrt{n}\langle n-1|$, $a|m\rangle = \sqrt{m}|m-1\rangle$, we get

$$A_U = -D(z)\Big(\sum_{n=1}^\infty \chi_{n,n-1}\sqrt{n}|n\rangle\langle n-1|\mathrm{d}z - \sum_{n=0}^\infty \chi_{n,n+1}\sqrt{n+1}|n\rangle\langle n+1|\mathrm{d}\bar{z}\Big)D^\dagger(z). \tag{40}$$

Changing the index by $n \rightarrow n+1$ in the first line and using the property $\chi_{n,n+1} = \chi_{n+1,n} = 1 - \operatorname{sech}\frac{\beta\hbar\omega}{2}$, the Uhlmann connection is finally expressed as

$$\begin{aligned}
A_U &= -\chi D(z)\left(a^\dagger \sum_{n=0}^\infty |n\rangle\langle n|\mathrm{d}z - \sum_{n=0}^\infty |n\rangle\langle n|a\mathrm{d}\bar{z}\right)D^\dagger(z) \\
&= -\chi\left[(a^\dagger - \bar{z})\mathrm{d}z - (a - z)\mathrm{d}\bar{z}\right],
\end{aligned} \tag{41}$$

where $\chi = 1 - \operatorname{sech}\frac{\beta\hbar\omega}{2}$ and Eq. (36) have been applied.

Let $g_C = \mathcal{P} e^{-\oint_C A_U}$ be the Uhlmann holonomy as the system traverses $C(t)$. In the Fock space spanned by $\{|n\rangle\}$, both $a$ and $a^\dagger$ are matrices of infinite dimensions, making it challenging to find an analytical expression of $g_C$. However, this can be achieved by solving the differential equation for $D(z)$. Using Eq. (36), it can be shown that Eq. (39) leads to a differential equation for $D(z(t))$. Explicitly,

$$\frac{\mathrm{d}D(z(t))}{\mathrm{d}t} = \left[ a^\dagger \dot{z} - a\dot{\bar{z}} - \frac{1}{2}(\bar{z}\dot{z} - z\dot{\bar{z}}) \right] D(z(t)), \tag{42}$$

as $z$ varies along the loop $C(t) = z(t)$. The solution to the above equation gives

$$D(z(t)) = \mathcal{P} e^{\int_0^t \{a^\dagger \dot{z}(t') - a\dot{\bar{z}}(t') - \frac{1}{2}[\bar{z}(t')\dot{z}(t') - z(t')\dot{\bar{z}}(t')]\}\mathrm{d}t'} D(z(0))$$
$$= e^{-\frac{1}{2}\int_0^t [\bar{z}(t')\dot{z}(t') - z(t')\dot{\bar{z}}(t')]\mathrm{d}t'} \mathcal{P} e^{\int_0^t [a^\dagger \dot{z}(t') - a\dot{\bar{z}}(t')]\mathrm{d}t'} D(z(0)). \tag{43}$$

Since $z(\tau) = z(0)$, $D(z(\tau)) = D(z(0))$ and it follows that

$$\mathcal{P} e^{\oint_C (a^\dagger \mathrm{d}z - a\mathrm{d}\bar{z})} = e^{\frac{1}{2}\oint_C (\bar{z}\mathrm{d}z - z\mathrm{d}\bar{z})} = e^{2iS_C}. \tag{44}$$

Here $S_C$ is the area enclosed by $C(t)$ along its counterclockwise direction. Let $\eta = 1 - \chi = \text{sech}\frac{\beta\hbar\omega}{2}$. The Uhlmann holonomy can be simplified as

$$g_C = \mathcal{P} e^{(1-\eta)\oint_C [(a^\dagger - \bar{z})\mathrm{d}z - (a-z)\mathrm{d}\bar{z}]} = e^{-2i(1-\eta^2)S_C} \mathbb{1}_\infty, \tag{45}$$

where $\mathbb{1}_\infty$ is the identity matrix in the bosonic Fock space, which is infinite-dimensional. Interestingly, although $g_C$ is generated by $A_U$, which belongs to an infinite-dimensional Lie algebra, it only forms a subgroup of U(1). Finally, the Uhlmann phase is given by

$$\theta_U = \arg \text{Tr}[\rho(z(0))g_C] = -2(1-\eta^2)S_C, \tag{46}$$

where Eq. (37) has been used.

In the zero-temperature limit, $\lim_{\beta\to\infty} \eta = 0$ and $\theta_U = -2S_C$, exactly agreeing with the Berry phase shown in Eq. (A.4). In the infinite-temperature limit, $\lim_{\beta\to 0} \eta = 1$, so $\theta_U = 0$ since $\rho(z(t))$ is always proportional to the identity operator in this case. While the physical meaning of the Uhlmann phase, especially the parallel-transport condition for $W$, awaits deeper explanations, the agreement of the Uhlmann phase with the Berry phase as $T \to 0$ in the case of infinite-dimensional bosonic coherent states offers more hints that their relation may be quite general.

## 3.2 Fermionic coherent states

Next, we verify if the Uhlmann phase approaches the Berry phase in fermionic coherent states, which may be constructed from the fermionic harmonic oscillator [35, 37]. We note that the Hamiltonian of a bosonic harmonic oscillator can be cast in the form $\hat{H} = \hbar\omega\{a^\dagger, a\}$. By considering the anticommutation relations of fermions versus the commutation relations of bosons, the Hamiltonian of a fermionic harmonic oscillator is $\hat{H} = \frac{\hbar\omega}{2}[b^\dagger, b] = \hbar\omega\left(b^\dagger b - \frac{1}{2}\right)$. Similar to its bosonic counterpart, the fermionic coherent state is also built via a translation to the vacuum:

$$|\xi\rangle = D(\xi)|0\rangle \equiv e^{b^\dagger \xi - \bar{\xi} b}|0\rangle. \tag{47}$$

Here $\xi$ is a Grassmann number and anticommutes with any fermionic operator. The translation operator $D(z)$ satisfies

$$D(\xi)bD^\dagger(\xi) = b - \xi, \quad D(\xi)b^\dagger D^\dagger(\xi) = b^\dagger - \bar{\xi}. \tag{48}$$

Similarly, parallel transport of a canonical ensemble of fermionic harmonic oscillators can be generated by a series of continuous translation by $D(\xi(t))$, where $\xi(t)$ is a closed curve of Grassmann numbers with $\xi(0) = \xi(\tau)$. The corresponding density matrix is

$$\rho(\xi(t)) = \frac{1}{Z} e^{-\beta D(\xi(t))\hat{H}D^\dagger(\xi(t))} = D(\xi(t))\rho(0)D^\dagger(\xi(t)), \tag{49}$$

where $\rho(0) = \frac{e^{-\beta\hat{H}}}{Z}$ with the partition function $Z = e^{\frac{1}{2}\beta\hbar\omega} + e^{-\frac{1}{2}\beta\hbar\omega} = 2\cosh\frac{\beta\hbar\omega}{2}$.

Since the system has a two-dimensional Hilbert space, the denominator of Eq. (32) is always $\lambda_0 + \lambda_1 = 1$. Consequently, the Uhlmann connection is simplified as

$$A_U = -\left[ d\sqrt{\rho(\xi)}, \sqrt{\rho(\xi)} \right]. \tag{50}$$

Let $\hat{N} = b^\dagger b$ be the number operator satisfying $\hat{N}^2 = \hat{N}$. It can be shown that

$$\rho(\xi) = \frac{1}{1 + e^{-\beta\hbar\omega}} - \tanh\left(\frac{\beta\hbar\omega}{2}\right)(b^\dagger - \bar{\xi})(b - \xi), \tag{51}$$

which further implies

$$d\sqrt{\rho(\xi)} = \frac{e^{\frac{1}{4}\beta\hbar\omega} - e^{-\frac{1}{4}\beta\hbar\omega}}{\sqrt{e^{\frac{1}{2}\beta\hbar\omega} + e^{-\frac{1}{2}\beta\hbar\omega}}}\left[ d\bar{\xi}(b - \xi) + (b^\dagger - \bar{\xi})d\xi \right]. \tag{52}$$

The Uhlmann connection then becomes

$$\begin{aligned}
A_U &= \frac{\left(e^{\frac{1}{4}\beta\hbar\omega} - e^{-\frac{1}{4}\beta\hbar\omega}\right)^2}{e^{\frac{1}{2}\beta\hbar\omega} + e^{-\frac{1}{2}\beta\hbar\omega}}\left[ d\bar{\xi}(b - \xi)(b^\dagger - \bar{\xi})(b - \xi) - (b^\dagger - \bar{\xi})(b - \xi)(b^\dagger - \bar{\xi})d\xi \right] \\
&= -\chi\left( b^\dagger d\xi - d\bar{\xi}b + d\bar{\xi}\xi - \bar{\xi}d\xi \right). \tag{53}
\end{aligned}$$

To evaluate the Uhlmann holonomy, we assume $\xi(t) = \zeta z(t)$, where $\zeta$ is a constant Grassmann number, and $z(t)$ $(0 \leq t \leq \tau)$ forms a closed curve $C$ in the $z$-plane. Thus, we have

$$g_C = \mathcal{P}e^{-\oint A_U} = e^{-4i\chi\bar{\zeta}\zeta S_C}\mathcal{P}e^{\chi\oint_C\left(b^\dagger\zeta dz - d\bar{z}\bar{\zeta}b\right)}. \tag{54}$$

Since the fermionic Fock space is only two-dimensional, the expression of $g_C$ of the fermionic coherent state can be directly evaluated without using the method of the bosonic coherent state. We expand the second term in the last line of Eq. (54) as

$$\begin{aligned}
\mathcal{P}e^{\chi\oint\left(b^\dagger\zeta dz - d\bar{z}\bar{\zeta}b\right)} &= 1 + \chi^2\int_0^\tau dt_1\int_0^{t_1} dt_2(b^\dagger\zeta\dot{z}_1 - \dot{\bar{z}}_1\bar{\zeta}b)(b^\dagger\zeta\dot{z}_2 - \dot{\bar{z}}_2\bar{\zeta}b) \\
&= 1 + \chi^2\int_0^\tau dt_1\int_0^{t_1} dt_2\bar{\zeta}\zeta\left(\dot{z}_1\dot{\bar{z}}_2 b^\dagger b - \dot{\bar{z}}_1\dot{z}_2 bb^\dagger\right). \tag{55}
\end{aligned}$$

where the first-order term vanishes due to $\oint dz = \oint d\bar{z} = 0$. $z_1 := z(t_1)$ and $z_2 := z(t_2)$ are introduced in the second-order term, and higher order terms vanish due to $\zeta^2 = \bar{\zeta}^2 = 0$ or $b^2 = b^{\dagger 2} = 0$. We evaluate the integral over $t_2$ and find the coefficient of $b^\dagger b$ becomes $\int_0^\tau dt_1\int_0^{t_1} dt_2\dot{z}_1\dot{\bar{z}}_2 = \int_0^\tau dt_1\dot{z}(t_1)[\bar{z}(t_1) - \bar{z}(0)] = \int_0^\tau dt_1\dot{z}(t_1)\bar{z}(t_1)$, where $\int_0^\tau dt_1\dot{z}(t_1)\bar{z}(0) = [z(\tau) - z(0)]\bar{z}(0) = 0$ has been applied. by the polar expression of $z$, $z(t) = r(t)e^{i\theta(t)}$, and substituting $\dot{z}\bar{z} = \dot{r}r + ir^2\dot{\theta}$ and $\dot{\bar{z}}z = \dot{r}r - ir^2\dot{\theta}$ into Eq. (55), the second term becomes

$$\chi^2\bar{\zeta}\zeta\int_0^\tau dt_1\left(\dot{z}\bar{z}b^\dagger b - \dot{\bar{z}}zbb^\dagger\right) = \frac{\chi^2\bar{\zeta}\zeta}{2}\int_{r(0)}^{r(\tau)}dr^2(b^\dagger b - bb^\dagger) + i\chi^2\bar{\zeta}\zeta r^2\int_0^{2\pi}d\theta(b^\dagger b + bb^\dagger)$$

$$= 2i\chi^2\bar{\zeta}\zeta S_C, \tag{56}$$

where we have applied $r(\tau) = r(0)$ and $S_C = \frac{1}{2}r^2\oint_C d\theta$. Once gain, by using $\bar{\zeta}^2 = \zeta^2 = 0$, the Uhlmann holonomy is given by

$$g_C = \left(1 - 4i\chi\bar{\zeta}\zeta S_C\right)\left(1 + 2i\chi^2\bar{\zeta}\zeta S_C\right)\mathbb{1}_2 = e^{-2i(1-\eta^2)\bar{\zeta}\zeta S_C}\mathbb{1}_2, \tag{57}$$

where $\mathbb{1}_2$ is the identity operator acting on the two-dimensional fermionic Fock space. With the help of Eq. (49), the Uhlmann phase of fermionic coherent state is

$$\theta_U = \arg\mathrm{Tr}\left[\rho(\xi(0))g_C\right] = -2(1-\eta^2)\bar{\zeta}\zeta S_C. \tag{58}$$

The expressions of both Uhlmann holonomy and Uhlmann phase are quite similar to their bosonic counterparts except the factor $\bar{\zeta}\zeta$, although they are obtained by different methods. Moreover, $\theta_U = 0$ as $T \to \infty$, and $\theta_U$ agrees with the Berry phase shown in Eq. (A.9 ) as $T \to 0$.

Interestingly, the results of both bosonic and fermionic coherent states exhibit an exact correspondence between the Uhlmann phase in the $T \to 0$ limit and the Berry phase. Although a full proof of the general case is challenging (see the next section), the results shown here and the previous results [22, 24, 26] all support the Uhlmann-Berry correspondence in the zero-temperature limit.

## 3.3 Additional example: Qutrit

After establishing the correspondence between the Uhlmann and Berry phases for both types of coherent states, here we conduct an extra check of the Uhlmann-Berry correspondence for a system with a finite-dimensional Hilbert space by examining the qutrit, a three-level system. A generalization of the Berry phase via the geometric phase for the generalized Bloch-sphere states of a three-level system has been discussed in Ref. [54]. The density matrix of a generic three-level system can be expanded by the identity matrix $\mathbb{1}_3$ and eight Gell-Mann matrices $\Lambda_i$ ($i = 1, 2\cdots, 8$), containing 8 controllable real parameters $\vec{n} = (n_1, n_2, \cdots, n_8)^T$. Explicitly, $\rho = \frac{1}{3}\left(\mathbb{1}_3 + \sqrt{3}\vec{n}\cdot\vec{\Lambda}\right)$, where $n_i = \frac{1}{2}\mathrm{Tr}(\rho\Lambda_i)$. The set $\mathbb{B}^8 = \{\vec{n} \in \mathbb{R}^8 | \vec{n}\cdot\vec{n} \le 1, \vec{n}^* = \vec{n}\}$ can be thought of as an eight-dimensional generalized Bloch sphere. When discussing the Uhlmann phase, a generic evolution path is a loop in $\mathbb{B}^8$, which has many possibilities. To present an exact correspondence between the Uhlmann and Berry phases, we instead simplify the qutrit model to a spin-$j$ paramagnet with $j = 1$, whose Uhlmann phase has been studied in Refs. [24, 25]. A loop in the parameter space of the spin-1 model corresponds to a loop on the two-dimensional unit sphere $S^2$. In the following, we verify the Uhlmann phase of the spin-1 model also reduces to the Berry phase as temperature approaches zero. We remark that the spin-1 system is topological [24] with a finite Hilbert space while the coheret states discussed previously are not topology but with infinite-dimensional Hilbert spaces.

Since the three components of the $j = 1$ angular momentum of a spin-1 paramagent can be spanned by the Gell-Mann matrices via $\hat{J}_x = \frac{1}{\sqrt{2}}\left(\Lambda_1 + \Lambda_6\right)$, $\hat{J}_y = \frac{1}{\sqrt{2}}\left(\Lambda_2 + \Lambda_7\right)$, and $\hat{J}_z = \frac{1}{2}\left(\Lambda_3 + \sqrt{3}\Lambda_8\right)$, the Hamiltonian of a spin-1 paramagnet in an external magnetic field can be expressed as $\hat{H} = \mu_B\mathbf{B}\cdot\hat{\mathbf{J}} = \mu_B\vec{d}\cdot\vec{\Lambda}$, where $\vec{d} = (\frac{B_x}{\sqrt{2}}, \frac{B_y}{\sqrt{2}}, \frac{B_z}{2}, 0, 0, \frac{B_x}{\sqrt{2}}, \frac{B_y}{\sqrt{2}}, \frac{\sqrt{3}B_z}{2})^T$, $\hbar\hat{\mathbf{J}}$ is the spin angular momentum of the particle, and $\mu_B$ is the Bohr magneton. The density matrix of the spin-$j$ paramagnet in canonical emsemble is $\rho = \frac{1}{Z}e^{-\beta\hat{H}}$. Therefore, the spin-1 model can be realized by a suitable choice of the parameter $(n_1, \cdots, n_8)^T$ of the original

qutrit model. The external magnetic field $\mathbf{B}$ can be parameterized by the polar and azimuthal angles $\theta, \phi$ as $\mathbf{B} = B(\sin\theta\cos\phi, \sin\theta\sin\phi, \cos\theta)^{\mathrm{T}}$. The Hamiltonian can be diagonalized as $\hat{H} = V(\theta, \phi)\omega_0 J_z V^\dagger(\theta, \phi)$, where $V(\theta, \phi) = \mathrm{e}^{-\mathrm{i}\phi J_z}\mathrm{e}^{-\mathrm{i}\theta J_y}\mathrm{e}^{\mathrm{i}\phi J_z}$. Thus, the eigenstates of $\hat{H}$ can be constructed as

$$|\psi_m^j(\theta, \phi)\rangle = \mathrm{e}^{-\mathrm{i}\phi(\frac{J_z}{\hbar}-m)}\mathrm{e}^{-\frac{\mathrm{i}}{\hbar}\theta J_y}|jm\rangle, \quad m = -j, -j+1, \cdots, j-1, j. \tag{59}$$

To simplify the notations, we adopt the natural unites such that $k_B = \hbar = 1$, and introduce $\omega_0 = \mu_B B$. A loop on $S^2$ can be expressed as $(\theta(t), \phi(t))$, and $V(\theta(t), \phi(t))$ actually defines an Uhlmann process if Uhlmann's parallel-transport condition is satisfied. By using

$$[\mathrm{d}\sqrt{\rho}, \sqrt{\rho}] = \frac{\{\mathrm{d}VV^\dagger, \mathrm{e}^{-\beta H}\}}{Z} + \frac{2\mathrm{e}^{-\frac{\beta H}{2}}V\mathrm{d}V^\dagger \mathrm{e}^{-\frac{\beta H}{2}}}{Z}, \tag{60}$$

it can be shown that the Uhlmann connection is

$$A_U = -\mathrm{i}\chi(J_x\sin\phi - J_y\cos\phi)\mathrm{d}\theta - \mathrm{i}\chi\left[(J_x\cos\phi + J_y\sin\phi)\cos\theta - J_z\sin\theta\right]\sin\theta\mathrm{d}\phi. \tag{61}$$

More details can be found in Ref. [24]. The Uhlmann phase depends on the path-ordered integral involving the matrix-valued $A_U$. If the evolution path is chosen as a circle of longitude or the equator (i.e., great circles), the exact expression of the path-ordered integral can be obtained. Interestingly, the Uhlmann phases for the two types of paths share the same expression:

$$\theta_U = \arg\sum_{m=-j}^{j}\frac{\mathrm{e}^{-\beta\omega_0 m}}{Z(0)}d_{mm}^j(2\pi\Omega\chi), \tag{62}$$

where $\chi = 1 - \mathrm{sech}(\beta\omega_0/2)$, $\Omega$ is the winding number and $d_{mm'}^j(\Theta) = \langle jm|\mathrm{e}^{-\mathrm{i}\Theta J_y}|jm'\rangle$ is the Wigner $d$-function. For $j = 1$, the explicit expression of the Uhlmann phase is

$$\theta_U = \arg\frac{1}{Z(0)}\left\{\cosh(\beta\omega_0)\left[1 + \cos\left(2\pi\Omega\mathrm{sech}\frac{\beta\omega_0}{2}\right)\right] + \cos\left(2\pi\Omega\mathrm{sech}\frac{\beta\omega_0}{2}\right)\right\}, \tag{63}$$

where $Z(0) = 1 + 2\cosh(\beta\omega_0)$. As $T \to 0$ or $\beta \to \infty$, $\mathrm{sech}\frac{\beta\omega_0}{2} = 0$, and $\theta_U = \arg 1 = 2\pi = 0 \ (mod \ 2\pi)$.

According to Eq. (59), the Berry phase of the $m$-th eigenstate along a loop $C(t)$ is evaluated as

$$\begin{aligned}
\theta_{Bm}(C) &= \mathrm{i}\int_0^\tau \mathrm{d}t\langle\psi_m^j|\frac{\mathrm{d}}{\mathrm{d}t}|\psi_m^j\rangle \\
&= \int_0^\tau \mathrm{d}t\langle jm|\left[-J_x\sin\theta\dot\phi + (J_z\cos\theta - m)\dot\varphi + J_y\dot\theta\right]|jm\rangle \\
&= -m\oint_C(1 - \cos\theta)\mathrm{d}\phi.
\end{aligned} \tag{64}$$

For the ground state of $j = 1$ along the equator, we have $m = -1$ and $\theta = \frac{\pi}{2}$. The Berry phase is then $\theta_{B-1} = 2\pi = 0 \ (mod \ 2\pi)$, which coincides with the value of $\theta_U$ as $T \to 0$. Therefore, the simplification of a qutrit to a spin-1 paramagnet offers another exactly solvable example of the correspondence between the Uhlmann and Berry phases.

# 4 Correspondence between Uhlmann phase and Berry phase

As shown in Sec. 2, the geometric frameworks of the Berry phase and Uhlmann phase are quite similar. The theory of the Uhlmann phase is built by following almost analogous steps as those of the Berry phase. They both start from the parallel-transport conditions, from which the corresponding Ehresmann connection $\omega$ is introduced to satisfy

$$
\begin{aligned}
\omega(\tilde{X}) &= 0, \quad \text{if } \tilde{X} \text{ is a horizontal vector,} \\
\omega(u^{\#}) &= u, \quad \text{if } u^{\#} \text{ is a vertical vector.}
\end{aligned}
\tag{65}
$$

The Berry and Uhlmann connections are the pullbacks of the corresponding $\omega$. This is why Uhlmann phase is a suitable generalization of the Berry phase to finite temperatures, at least from the point of view of geometry. The comparison leads to the question on whether the Uhlmann phase always reduces to the Berry phase as $T \to 0$.

In the following, a conditional proof will be constructed in a progressive manner. Firstly, we point out a class of special case that should not be considered in the correspondence by noting that the theory of the Uhlmann phase is built on the assumption that the density matrix must be full rank, which excludes pure states if the dimension of the Hilbert space is larger than one. Therefore, systems with a 1D Hilbert space should be treated as special cases because there is no sensible meaning of thermal distribution, as the system has no other states to distribute the weight. In Appendix B, we show the Uhlmann connection vanishes identically for systems with a 1D Hilbert space, leading to a vanishing Uhlmann phase for those special cases. In contrast, the Berry phase of a system with a 1D Hilbert space needs not vanish since a pure state may be considered as a 1D Hilbert space during an adiabatic evolution.

For the more general cases, it has been reported that the Uhlmann phase indeed approaches the Berry phase as $T \to 0$ for two-level and four-level systems [22, 26]. We already demonstrated that the spin-1 system supports the correspondence, and one may verify this is the case for generic spin-$j$ paramagnets in magnetic fields by following Refs. [24, 25]. However, it has not been proven if the correspondence between the Uhlmann and Berry phases is a general conclusion since at first look, the expressions of the Berry phase and the Uhlmann phase are in general different. If the question has a positive answer, it will provide a correspondence between the geometric phases of pure and mixed states even though the underlying bundles are very different, in the sense that the fiber bundle associated with the Berry phase may be nontrivial while that associated with the Uhlmann bundle is always trivial [46]. Thus, the correspondence cannot be at the level of the underlying bundles.

To understand the correspondence between the Berry and Uhlmann phases, we analyze the Uhlmann connection (35) and search for any relation to the Berry connection. We assume the quantum system is in a thermal-equilibrium state at temperature $T$ with $\rho = \frac{e^{-\beta \hat{H}}}{Z}$, where $Z$ is the partition function. Since $\rho = \sum_n \lambda_n |n\rangle\langle n|$ and $\hat{H}$ share the eigenvectors, we assume $\hat{H}|n\rangle = E_n|n\rangle$. Furthermore, we will write $|n\rangle \equiv |E_n\rangle$ in the following and only consider the case without energy degeneracy for simplicity. Let $E_0 < E_1 < \cdots$, then

$$
\lim_{T \to 0} \frac{\lambda_n}{\lambda_m} = \lim_{\beta \to \infty} e^{-\beta(E_n - E_m)} = 0, \quad \text{if } n > m.
\tag{66}
$$

Note that $\lambda_n \neq \lambda_m$ in Eq. (35). Thus, we set $\lambda_{\min} = \min\{\lambda_n, \lambda_m\}$ and $\lambda_{\max} = \max\{\lambda_n, \lambda_m\}$. This implies

$$
\lim_{T \to 0} \frac{\left(\sqrt{\lambda_n} - \sqrt{\lambda_m}\right)^2}{\lambda_n + \lambda_m} = \lim_{T \to 0} \frac{\left(1 - \sqrt{\frac{\lambda_{\min}}{\lambda_{\max}}}\right)^2}{1 + \frac{\lambda_{\min}}{\lambda_{\max}}} = 1.
\tag{67}
$$

The Uhlmann connection (35) in the zero-temperature limit then becomes

$$
\begin{aligned}
A_U &\to -\sum_{n\neq m}|n\rangle\langle n|\mathrm{d}|m\rangle\langle m| \\
&= -\sum_{nm}|n\rangle\langle n|\mathrm{d}|m\rangle\langle m| + \sum_n |n\rangle\langle n|\mathrm{d}|n\rangle\langle n| \\
&= -\sum_n \mathrm{d}|n\rangle\langle n| + \sum_n \langle n|\mathrm{d}|n\rangle|n\rangle\langle n|.
\end{aligned}
\tag{68}
$$

Interestingly, the second term of $A_U$ is the Berry connection for each energy level. When evaluating $\theta_U$ by Eq. (28), every step must be treated carefully. We emphasize that the trace must be taken after evaluating the path-ordered integral since the path-ordering and Taylor-expansion operations may not commute with each other. Moreover, the path-ordered integrals themselves are also challenging. For example, when dealing with $\theta_U$ of bosonic coherent states in the previous section, we have developed a technique to handle the difficulties. In some other situations [22, 24, 25], $A_U$ may be proportional to a constant matrix when the system follows a special path in the parameter space, thereby making the the path-ordering operator $\mathcal{P}$ manageable. However, those cases depend on the details of the loop $C(t)$ and even the specific coordinates chosen to evaluate $A_U$, so they are not easy to be generalized to generic systems. The challenge of evaluating the Uhlmann phase is somewhat similar to the difficulties in dealing with the time-ordering operation in quantum field theory, where techniques like the Feynman diagrams have been developed to facilitate a perturbative expansion [33, 55].

Nevertheless, a conditional proof can be obtained to show that the Uhlmann phase indeed approaches the Berry phase in the zero-temperature limit. An examination the bosonic and fermionic coherent states discussed previously reveals two important features: (1) The Uhlmann and Berry phases are both generated by unitary processes, and (2) the Berry connection of each energy level has the same expression, as indicated by Eqs. (A.2) and (A.7). Here the unitary Uhlmann process means the density matrix follows Eq. (37) with $z = z(t)$, and the eigen-energies $E_n$'s remain unchanged during the process. Hence, we consider a class of unitary Uhlmann processes characterized by those two features. When the parameter takes the value $t$, each energy level satisfies $|n(t)\rangle = \mathcal{D}(t)|n(0)\rangle$ with an unitary operator $\mathcal{D}(t)$ satisfying the cyclic condition $\mathcal{D}(\tau) = 1$. The Berry connection for each level is assumed the same:

$$
A_B = \langle n(t)|\mathrm{d}|n(t)\rangle = \langle n(0)|\mathcal{D}^\dagger \mathrm{d}\mathcal{D}|n(0)\rangle.
\tag{69}
$$

According to Eq. (68), in the $T \to 0$ limit, the Uhlmann connection is

$$
\lim_{T\to 0} A_U = -\sum_n \mathrm{d}|n(t)\rangle\langle n(t)| + \sum_n \langle n(t)|\mathrm{d}|n(t)\rangle|n(t)\rangle\langle n(t)| = A_B - \mathrm{d}\mathcal{D}\mathcal{D}^{-1},
\tag{70}
$$

where the completeness of the instantaneous energy eigenstates has been applied. Interestingly, Eq. (70) indicates that the Uhlmann and Berry connections are off by a gauge transformation, which actually renders no contribution after a contour integral along a closed loop. Explicitly, $\oint \mathrm{d}\mathcal{D}\mathcal{D}^{-1} = \oint \mathrm{d}\ln\mathcal{D} = 0$. Hence, the Uhlmann phase in the zero-temperature limit is given by

$$
\lim_{T\to 0} \theta_U = \arg \mathrm{Tr}[\rho(0)\mathcal{P}e^{-\oint A_U}] = \arg\left\{\mathrm{Tr}[\rho(0)]e^{-\oint A_B}\right\} = \theta_B,
\tag{71}
$$

where Eq. (70) has been used, and the path-ordering is dropped in the second line since $A_B \in \mathfrak{u}(1)$.

Importantly, the two conditions of the previous proof may be relaxed or changed further. Firstly, the condition that the Uhlmann process is unitary can be dropped. We recall the generic

expression (68) and introduce the unitary transformation $\mathcal{D}(t) = \sum_n |n(t)\rangle\langle n(0)|$ satisfying $|n(t)\rangle = \mathcal{D}(t)|n(0)\rangle$. Although $\mathcal{D}^\dagger\mathcal{D} = \mathcal{D}\mathcal{D}^\dagger = 1$, it does not necessarily imply the corresponding physical process is unitary since the condition $E_n(t) = E_n(0)$ may not be guaranteed during the process. Therefore, the density matrix does not necessarily obey the transformation (37). Moreover, the condition that the Berry connection of each level is the same can be replaced by introducing the Berry connection matrix:

$$\hat{A}_B = \sum_n A_{Bn}|n(t)\rangle\langle n(t)| = \sum_n \langle n(t)|\mathrm{d}|n(t)\rangle|n(t)\rangle\langle n(t)|. \tag{72}$$

In this more general case, it can be shown that $A_U = \hat{A}_B - \mathrm{d}\mathcal{D}\mathcal{D}^{-1}$. Once again, we have $\oint \mathrm{d}\mathcal{D}\mathcal{D}^{-1} = 0$. According to Eq. (66), the weight factor of the ground state is infinitely larger than that of any excited state when $T \to 0$, i.e., $\lambda_0 = \frac{e^{-\beta E_0}}{Z} \approx 1$. Thus, the initial density matrix can be reasonably approximated as

$$\rho(0) \approx |E_0(0)\rangle\langle E_0(0)|, \tag{73}$$

and the Uhlmann phase is then

$$\lim_{T\to 0} \theta_U = \arg\langle E_0(0)|\mathcal{P}e^{-\oint A_U}|E_0(0)\rangle = \arg\langle E_0(0)|\mathcal{P}e^{-\oint \hat{A}_B}|E_0(0)\rangle. \tag{74}$$

Since $\hat{A}_B \in u(N)$, the path-ordering operation $\mathcal{P}$ is nontrivial in general. Therefore, we need to add a condition here. When $\hat{A}_B$ is a diagonal matrix in the space spanned by $\{|n(0)\rangle\}$ or a constant matrix as the system traverses a specific loop in the parameter space, the path-ordering operation $\mathcal{P}$ is trivial, and the integrals can be carried out. In those situations, the Uhlmann phase becomes

$$\begin{aligned}\lim_{T\to 0} \theta_U &= \arg\langle E_0(0)|e^{-\oint \sum_n A_{Bn}|n(t)\rangle\langle n(t)|}|E_0(0)\rangle\\ &= \arg\left[|\langle E_0(0)|E_n(t)\rangle|^2\langle E_0(0)|e^{-\oint \sum_n A_{Bn}|n(0)\rangle\langle n(0)|}|E_0(0)\rangle\right]\\ &= \theta_{B0}, \end{aligned} \tag{75}$$

where $|E_n(t)\rangle = |n(t)\rangle$ has been used, and $\theta_{B0}$ is the Berry phase of the ground state. The proof of the correspondence between the Berry and Uhlmann phases is already quite general although we still need the relaxed assumption of the form of $\hat{A}_B$. We expect the most general proof, which still needs to exclude the special cases with a 1D Hilbert space, will be completed in future research of the Uhlmann phase.

## 5 Experimental implications

Since bosonic coherent states play a fundamental role in quantum optics [39–41], we discuss possible experimental realizations and measurements of the Uhlmann phase of bosonic coherent states. We first outline the basic ideas for constructing a protocol and leave the detailed techniques for future studies. There have been many ways to realize and manipulate coherent states by various experimental strategies [56–60], which may be implemented to fill in the necessary steps for the experimental demonstration of the Uhlmann phase of many-body systems, exemplified by the bosonic coherent states.

There are two important issues that need to be addressed in the protocol. The first is to suitably represent a mixed state, which can be characterized by the purification or purified state of a density matrix. The purification $W = \sqrt{\rho}U$ is not necessarily a Hermitian matrix,

and its physical interpretation is still under debate. With the advancement of quantum computation, the purified state $|W\rangle$ has become realizable [24, 61]. Therefore, the purified state is a more viable way for physical realizations. Explicitly, one can construct an entangled state between the system of interest and an ancilla encoding the environmental effects in the form $|W\rangle = \sum_n \sqrt{\lambda_n}|n\rangle_s \otimes U^T|n\rangle_a$. Here the subscripts $s$ and $a$ respectively represent the system and ancilla. The thermal distribution determines the coefficients while the $U(N)$ factor acts on the ancilla.

The second issue is to design a proper physical process for simulating the parallel transport of $|W\rangle$ and generating the correct Uhlmann process. This is complicated by the fact that $|W\rangle$ is formally a state vector and cannot satisfy a matrix-valued equation that is fully equivalent to the condition (18). A solution [24] is to follow the condition (17) to perform parallel transport of the state. An explicit construction is as follows. An Uhlmann process can be generated by controlling the parameter $z$, which forms a closed curve $C(t) = z(t)$ $(0 \le t \le \tau)$ in the parameter space, which in this case is the complex plane. Thus, the density matrix of a bosonic coherent-state harmonic oscillator is given by

$$\rho(z(t)) = \sum_{n=0}^{\infty} \lambda_n |n, z(t)\rangle\langle n, z(t)| = \sum_{n=0}^{\infty} \lambda_n D(z(t))|n\rangle\langle n|D^\dagger(z(t)), \tag{76}$$

where $\lambda_n = \frac{1}{Z}e^{-\beta\hbar\omega(n+\frac{1}{2})}$ is independent of $t$ since $D(z(t))$ is a unitary transformation. The corresponding purified state is given by

$$|W(z(t))\rangle = \sum_{n=0}^{\infty} \sqrt{\lambda_n} D(z(t))|n\rangle_s \otimes D^*(z(t))|n\rangle_a, \tag{77}$$

where $D^* = (D^\dagger)^T$ has been applied. However, we emphasize there is a subtlety about the transpose [24, 62], as the purified state needs to satisfy the Hilbert-Schmidt inner product (16). Moreover, Eq. (39) leads to an equivalent identity:

$$[dD(z)]D^\dagger(z) = \left(a^\dagger - \frac{1}{2}\bar{z}\right)dz - \left(a - \frac{1}{2}z\right)d\bar{z}. \tag{78}$$

Using these two equations and

$$\frac{d}{dt}|W(z(t))\rangle = \sum_{n=0}^{\infty} \sqrt{\lambda_n}(\dot{D} \otimes D^* + D \otimes \dot{D}^*)|n\rangle_s \otimes |n\rangle_a, \tag{79}$$

the weakened parallel-transport condition (17) can be verified straightforwardly:

$$\langle W(z(t))|\frac{d}{dt}|W(z(t))\rangle = \sum_{n=0}^{\infty} \lambda_n \left({}_s\langle n|D^\dagger \dot{D}|n\rangle_s + {}_a\langle n|\dot{D}D^\dagger|n\rangle_a\right) = 0. \tag{80}$$

Therefore, if the purified state follows the parallel-transport condition of Eq. (80), the evolution simulates an Uhlmann cycle as $t$ goes from 0 to $\tau$. The Uhlmann phase is then given by the phase difference between the initial and final purified states:

$$\theta_U = \arg\langle W(0)|W(\tau)\rangle. \tag{81}$$

The expression now involves (I) the transition amplitude between the initial and final purified states and (II) phase extraction. For bosonic coherent states, the former may be realized by entangling two coherent states with one acting as the system and the other as the ancilla and evolve the system according to the parallel-transport condition. The latter may be performed by interferometric or tomographic means on the overlap between the initial and final purified states. Since a bosonic coherent state is a many-body state involving infinite particles, both tasks are challenging and await future experimental realizations.

# 6 Conclusion

Through the bundle language, we concisely show the analogous frameworks of the Berry phase and Uhlmann phase via the concepts of parallel transport and holonomy. As concrete examples, we present the analytic expressions of the Uhlmann phases of bosonic and fermionic coherent states and reveal the geometric information carried by them. In addition to the smooth dependence on temperature, the Uhlmann phases of both cases approach their corresponding Berry phases as $T \to 0$, providing another set of exactly solvable examples supporting the agreement between the Uhlmann and Berry phases in the zero-temperature limit. Except special cases like those with a 1D Hilbert space, we propose that the correspondence between the Uhlmann and Berry phases is a general property of quantum systems. The conditional proof of the correspondence lays the foundation for a complete proof in the future and provides more insights into the relations between pure and mixed states.

**Funding information**   H. G. was supported by the National Natural Science Foundation of China (Grant No. 12074064). C. C. C. was supported by the National Science Foundation under Grant No. PHY-2011360.

# A  Berry phase of coherent states

## A.1  Bosonic coherent state

We assume the state $|n, z\rangle = D(z)|n\rangle$ of a bosonic harmonic oscillator evolves adiabatically along the curve $C(t) := z(t)$ with $z(0) = z(\tau)$ ($0 \le t \le \tau$). The Berry phase generated during the evolution is

$$\theta_{Bn}(C) = i \int_{C,0}^{\tau} dt \langle n, z(t)| \frac{\partial}{\partial t} |n, z(t)\rangle = i \oint_C d\mathbf{x} \cdot \langle n|D^\dagger(z)\nabla D(z)|n\rangle, \qquad (A.1)$$

where $\nabla = \vec{e}_x \frac{\partial}{\partial x} + \vec{e}_y \frac{\partial}{\partial y}$ is the gradient at the point $z = x + iy$. The Berry connection is given by

$$A_{Bn} = \langle n|D^\dagger(z)dD(z)|n\rangle = \frac{1}{2}(\bar{z}dz - zd\bar{z}). \qquad (A.2)$$

Further calculations show that

$$D^\dagger(z)\frac{\partial D(z)}{\partial x} = -iy + (a^\dagger - a), \quad D^\dagger(z)\frac{\partial D(z)}{\partial y} = ix + i(a^\dagger + a). \qquad (A.3)$$

Therefore,

$$\theta_{Bn}(C) = \oint_C (ydx - xdy) = \frac{i}{2}\oint_C (\bar{z}dz - zd\bar{z}) = -2S_C. \qquad (A.4)$$

We emphasize that the contour integral is evaluated along the counterclockwise direction of $C(t)$.

## A.2  Fermionic coherent state

Similarly, the Berry connection of the fermionic coherent state $|n, \xi\rangle = D(\xi)|n\rangle$ is

$$A_{Bn} = \langle n, \xi|d|n, \xi\rangle = \langle n|D^\dagger(\xi)dD(\xi)|n\rangle. \qquad (A.5)$$

Using

$$D^\dagger(\xi)\mathrm{d}D(\xi) = \left(b^\dagger + \frac{1}{2}\bar{\xi}\right)\mathrm{d}\xi + \left(b + \frac{1}{2}\xi\right)\mathrm{d}\bar{\xi}, \tag{A.6}$$

we get

$$A_{Bn} = \frac{1}{2}\left(\bar{\xi}\mathrm{d}\xi + \xi\mathrm{d}\bar{\xi}\right) = \frac{1}{2}\left(\bar{\xi}\mathrm{d}\xi - \mathrm{d}\bar{\xi}\xi\right), \tag{A.7}$$

which has a similar expression to its bosonic counterpart (A.2 ). Substituting $\xi = \zeta z$, where $\zeta$ is a constant Grassmann number, into Eq. (A.7 ), it becomes

$$A_{Bn} = \frac{1}{2}\bar{\zeta}\zeta\left(\bar{z}\mathrm{d}z - z\mathrm{d}\bar{z}\right). \tag{A.8}$$

The Berry phase is

$$\theta_{Bn} = \mathrm{i}\oint_C A_{Bn} = -2\bar{\zeta}\zeta S_C. \tag{A.9}$$

The expression is also similar to the Berry phase of bosonic coherent states except the factor of $\bar{\zeta}\zeta$.

## B    Uhlmann connection in 1D Hilbert space

If we consider the density matrix of a 1D Hilbert space, $\rho(t) = |\psi(t)\rangle\langle\psi(t)|$, it is straightforward to show that

$$[\mathrm{d}\sqrt{\rho}, \sqrt{\rho}] = \mathrm{d}|\psi\rangle\langle\psi| + |\psi\rangle(\mathrm{d}\langle\psi|)|\psi\rangle\langle\psi| - |\psi\rangle\mathrm{d}\langle\psi| - |\psi\rangle\langle\psi|(\mathrm{d}|\psi\rangle)\langle\psi|. \tag{B.1}$$

Thus, $\langle\psi|[\mathrm{d}\sqrt{\rho}, \sqrt{\rho}]|\psi\rangle = 0$, which leads to $A_U = 0$. This also implies that the Uhlmann connection and Uhlmann phase of a pure state are always zero. In contrast, the corresponding Berry connection after a cyclic adiabatic process, $A_B = \langle\mathbf{R}|\mathrm{d}|\mathbf{R}\rangle$, and the Berry phase $\theta_B = \mathrm{i}\oint A_B$ is not zero in general. Here $|\psi(t)\rangle = \mathrm{e}^{-\int_0^t \langle\mathbf{R}(t')|\frac{\mathrm{d}}{\mathrm{d}t'}|\mathbf{R}(t')\rangle\mathrm{d}t'}|\mathbf{R}(t)\rangle$. Since a single pure state is equivalent to a system in a 1D Hilbert space and may accumulate a nontrivial Berry phase, the Uhlmann phase does not reduce to the Berry phase as $T \to 0$ in this type of special cases.

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
