# Peer review of "Uhlmann phase of coherent states and the Uhlmann-Berry correspondence"

_SciPost Physics Core, doi:SciPost Phys. Core 6, 024 (2023)_

## Round 1 · Referee Report · Anonymous · 2022-11-5

Strengths
1. The motivation is consistent with the proposed problem.
2. The article is clear, and the construction of the Uhlmann connection is interesting and correct.
Weaknesses
1. This problem has already been attacked in various ways, for example, Phys.Rev.Lett.85(2000)2845, Phys.Rev.Lett.91(2003)090405, Phys.Rev.Lett.94(2005). )050401, Phys.Rev.A73(2006)012107, and some more. The authors of these articles essentially obtained the same result, which is why it should be clarified in much greater depth in what sense the results obtained are worthy of publication compared with the previous results.
2. The examples are not very interesting since they do not reveal the usefulness of the introduced formalism in depth.
Report
The article is generally well written; however, given the wide variety of results in this line, it needs to be completely rewritten to account for previous developments and clearly show how its formalism is worth publishing. In addition, the examples considered do not reveal the usefulness of their formalism and could easily be solved with the previous formalisms, so it is necessary to consider some systems like the one reported in Phys.Rev.A82(2010)062108.
Chih-Chun Chien on 2022-11-16 [id 3034]
We thank the referee for reviewing our manuscript. We are willing to revise the manuscript to address the comments. Specifically,
“1. This problem has already been attacked in various ways, for example, Phys.Rev.Lett.85(2000)2845, Phys.Rev.Lett.91(2003)090405, Phys.Rev.Lett.94(2005). )050401, Phys.Rev.A73(2006)012107, and some more. The authors of these articles essentially obtained the same result, which is why it should be clarified in much greater depth in what sense the results obtained are worthy of publication compared with the previous results.” Moreover, the referee comments, “it needs to be completely rewritten to account for previous developments and clearly show how its formalism is worth publishing”.
We thank the referee for mentioning more previous works on generalizing the Berry phase of pure states to mixed states. Nevertheless, the Uhlmann phase implemented in our work differs from other works in one important aspect: The Uhlmann phase is constructed from the Uhlmann bundle of density matrices, which gives the framework a completely geometric construction and concrete physical meaning. Through the explicit construction, all geometric quantities from the Berry bundle of purified states find their counterparts in the Uhlmann bundle. Moreover, the Uhlmann phase from the Uhlmann connection has been shown to exhibit quantization and finite-temperature topological phase transitions in previous works.
However, two important questions remain unanswered for the Uhlmann phase: (1) All previous examples only deal with systems with finite-dimensional Hilbert spaces, and it has not been shown if the Uhlmann phase applies to systems with infinite-dimensional Hilbert spaces. (2) The Uhlmann bundle is different from the Berry bundle in the sense that the former is a trivial bundle but the latter need not be trivial. Meanwhile, the Uhlmann phase seems to approach the Berry phase in the zero-temperature limit. Our manuscript addresses both questions by first presenting the explicit expressions of the Uhlmann phases of the bosonic and fermionic coherent states, which serve as explicit examples of systems with infinite-dimensional Hilbert spaces. Then we clarify the correspondence between the Uhlmann and Berry phase and outline a conditional proof to show that their correspondence is not at the level of the bundles or connections but at the level of the holonomies.
To address the comments from the referee, we are willing to add a paragraph summarizing all the previous works mentioned by the referee and present a fair comparison of various formalisms to help the reader appreciate their differences from the Uhlmann-phase approach. We will also rewrite the descriptions about the Uhlmann phase to make the two main points more transparent to help the reader grasp the central message.
“2. The examples are not very interesting since they do not reveal the usefulness of the introduced formalism in depth.” Moreover, the referee comments, “it is necessary to consider some systems like the one reported in Phys.Rev.A82(2010)062108.”
We respectfully disagree with the referee regarding the examples of our manuscript. While previous works have shown the Uhlmann phases of two-level systems, spin-j systems, Chern insulator, time-reversal invariant topological insulator, to name a few, all of them are systems with finite-dimensional Hilbert spaces. In contrast, the bosonic and fermionic coherent states are examples with infinite-dimensional Hilbert spaces, which are more challenging and their solutions will show that the Uhlmann phase is indeed a universal framework. Moreover, previous works have shown the Uhlmann phase approaches the Berry phase in the zero-temperature limit only for systems with finite-dimensional Hilbert spaces. Our examples of the coherent states complete the demonstration that the Uhlmann phase approaches the Berry phase in general.
Nevertheless, we are willing to include the example mentioned by the referee, which is a three-level system. We will derive the full expression of the Uhlmann phase of the system and show its Uhlmann-Berry correspondence. Interestingly, a three-level system is similar to a spin-1 system, and our previous work (Phys. Rev. A 104, 023303 (2021), cited as [24]) already shows the quantization of the Uhlmann phase of a spin-1 system and its correspondence with the Berry phase. The inclusion of the example mentioned by the referee will help the reader appreciate the generality of the Uhlmann-Berry correspondence.

---

## Round 2 · Author Response

We thank the previous referee for reviewing our manuscript. We have revised our manuscript to address the comments. Specifically,
“1. This problem has already been attacked in various ways, for example, Phys.Rev.Lett.85(2000)2845, Phys.Rev.Lett.91(2003)090405, Phys.Rev.Lett.94(2005). )050401, Phys.Rev.A73(2006)012107, and some more. The authors of these articles essentially obtained the same result, which is why it should be clarified in much greater depth in what sense the results obtained are worthy of publication compared with the previous results.” Moreover, the referee comments, “it needs to be completely rewritten to account for previous developments and clearly show how its formalism is worth publishing”.
We thank the referee for mentioning more previous works on generalizing the Berry phase of pure states to mixed states. Nevertheless, the Uhlmann phase implemented in our work differs from other works in one important aspect: The Uhlmann phase is constructed from the Uhlmann holonomy of the Uhlmann bundle of density matrices, which gives the framework a completely geometric construction and concrete physical meaning. Through the explicit construction, all geometric quantities from the Berry bundle of purified states find their counterparts in the Uhlmann bundle. Moreover, the Uhlmann phase from the Uhlmann connection has been shown to exhibit quantization and finite-temperature topological phase transitions in previous works.
However, two important questions remain unanswered for the Uhlmann phase: (1) All previous examples only deal with systems with finite-dimensional Hilbert spaces, and it has not been shown if the Uhlmann phase applies to systems with infinite-dimensional Hilbert spaces. (2) The Uhlmann bundle is different from the Berry bundle in the sense that the former is a trivial bundle but the latter need not be trivial. Meanwhile, the Uhlmann phase seems to approach the Berry phase in the zero-temperature limit. Our manuscript addresses both questions by first presenting the explicit expressions of the Uhlmann phases of the bosonic and fermionic coherent states, which serve as explicit examples of systems with infinite-dimensional Hilbert spaces. Then we clarify the correspondence between the Uhlmann and Berry phase and outline a conditional proof to show that their correspondence is not at the level of the bundles or connections but at the level of the holonomies.
To address the comments from the referee, we have added a paragraph in the Introduction summarizing all the previous works mentioned by the referee and present a fair comparison of other formalisms to help the reader appreciate their differences from the Uhlmann-phase approach.
“2. The examples are not very interesting since they do not reveal the usefulness of the introduced formalism in depth.” Moreover, the referee comments, “it is necessary to consider some systems like the one reported in Phys.Rev.A82(2010)062108.”
We respectfully disagree with the referee regarding the examples of our manuscript. While previous works have shown the Uhlmann phases of two-level systems, spin-j systems, Chern insulator, time-reversal invariant topological insulator, to name a few, all of them are systems with finite-dimensional Hilbert spaces. In contrast, the bosonic and fermionic coherent states are examples with infinite-dimensional Hilbert spaces, which are more challenging and their solutions will show that the Uhlmann phase is indeed a universal framework. Moreover, previous works have shown the Uhlmann phase approaches the Berry phase in the zero-temperature limit only for systems with finite-dimensional Hilbert spaces. Our examples of the coherent states with infinite-dimensional Hilbert spaces complete the demonstration that the Uhlmann phase approaches the Berry phase in general.
Nevertheless, we have added the example mentioned by the referee, which is a three-level system. Since a three-level system is too general, we simplify the model to be equivalent to a spin-1 system. From our previous work (Phys. Rev. A 104, 023303 (2021), cited as [24]), we show the explicit expression of the Uhlmann phase of the simplified three-state system and the exact correspondence with the Berry phase as the temperature goes to zero. The inclusion of the example mentioned by the referee will help the reader appreciate the generality of the Uhlmann-Berry correspondence.
“1. This problem has already been attacked in various ways, for example, Phys.Rev.Lett.85(2000)2845, Phys.Rev.Lett.91(2003)090405, Phys.Rev.Lett.94(2005). )050401, Phys.Rev.A73(2006)012107, and some more. The authors of these articles essentially obtained the same result, which is why it should be clarified in much greater depth in what sense the results obtained are worthy of publication compared with the previous results.” Moreover, the referee comments, “it needs to be completely rewritten to account for previous developments and clearly show how its formalism is worth publishing”.
We thank the referee for mentioning more previous works on generalizing the Berry phase of pure states to mixed states. Nevertheless, the Uhlmann phase implemented in our work differs from other works in one important aspect: The Uhlmann phase is constructed from the Uhlmann holonomy of the Uhlmann bundle of density matrices, which gives the framework a completely geometric construction and concrete physical meaning. Through the explicit construction, all geometric quantities from the Berry bundle of purified states find their counterparts in the Uhlmann bundle. Moreover, the Uhlmann phase from the Uhlmann connection has been shown to exhibit quantization and finite-temperature topological phase transitions in previous works.
However, two important questions remain unanswered for the Uhlmann phase: (1) All previous examples only deal with systems with finite-dimensional Hilbert spaces, and it has not been shown if the Uhlmann phase applies to systems with infinite-dimensional Hilbert spaces. (2) The Uhlmann bundle is different from the Berry bundle in the sense that the former is a trivial bundle but the latter need not be trivial. Meanwhile, the Uhlmann phase seems to approach the Berry phase in the zero-temperature limit. Our manuscript addresses both questions by first presenting the explicit expressions of the Uhlmann phases of the bosonic and fermionic coherent states, which serve as explicit examples of systems with infinite-dimensional Hilbert spaces. Then we clarify the correspondence between the Uhlmann and Berry phase and outline a conditional proof to show that their correspondence is not at the level of the bundles or connections but at the level of the holonomies.
To address the comments from the referee, we have added a paragraph in the Introduction summarizing all the previous works mentioned by the referee and present a fair comparison of other formalisms to help the reader appreciate their differences from the Uhlmann-phase approach.
“2. The examples are not very interesting since they do not reveal the usefulness of the introduced formalism in depth.” Moreover, the referee comments, “it is necessary to consider some systems like the one reported in Phys.Rev.A82(2010)062108.”
We respectfully disagree with the referee regarding the examples of our manuscript. While previous works have shown the Uhlmann phases of two-level systems, spin-j systems, Chern insulator, time-reversal invariant topological insulator, to name a few, all of them are systems with finite-dimensional Hilbert spaces. In contrast, the bosonic and fermionic coherent states are examples with infinite-dimensional Hilbert spaces, which are more challenging and their solutions will show that the Uhlmann phase is indeed a universal framework. Moreover, previous works have shown the Uhlmann phase approaches the Berry phase in the zero-temperature limit only for systems with finite-dimensional Hilbert spaces. Our examples of the coherent states with infinite-dimensional Hilbert spaces complete the demonstration that the Uhlmann phase approaches the Berry phase in general.
Nevertheless, we have added the example mentioned by the referee, which is a three-level system. Since a three-level system is too general, we simplify the model to be equivalent to a spin-1 system. From our previous work (Phys. Rev. A 104, 023303 (2021), cited as [24]), we show the explicit expression of the Uhlmann phase of the simplified three-state system and the exact correspondence with the Berry phase as the temperature goes to zero. The inclusion of the example mentioned by the referee will help the reader appreciate the generality of the Uhlmann-Berry correspondence.

---

## Round 2 · List of Changes

- A new paragraph comparing the Uhlmann phase to other geometric phases of mixed states has been added to the Introduction. All the references mentioned by the referee have been cited in the revised version.
- A new example of the Uhlmann phase of a three-level system and its correspondence to the Berry phase has been added to the new Section 3.3.
- Some typos have been corrected.

---

## Editorial Decision

published